# Retrosynthesis of multi-component metal–organic frameworks

Shuai Yuan[1], Jun-Sheng Qin[1], Jialuo Li[1], Lan Huang [2], Liang Feng[1], Yu Fang[1], Christina Lollar[1], Jiandong Pang[1], Liangliang Zhang[1], Di Sun [3], Ali Alsalme [4], Tahir Cagin[2,5] & Hong-Cai Zhou [1,2,4]

Crystal engineering of metal–organic frameworks (MOFs) has allowed the construction of complex structures at atomic precision, but has yet to reach the same level of sophistication as organic synthesis. The synthesis of complex MOFs with multiple organic and/or inorganic components is ultimately limited by the lack of control over framework assembly in one-pot reactions. Herein, we demonstrate that multi-component MOFs with unprecedented complexity can be constructed in a predictable and stepwise manner under simple kinetic guidance, which conceptually mimics the retrosynthetic approach utilized to construct complicated organic molecules. Four multi-component MOFs were synthesized by the subsequent incorporation of organic linkers and inorganic clusters into the cavity of a mesoporous MOF, each composed of up to three different metals and two different linkers. Furthermore, we demonstrated the utility of such a retrosynthetic design through the construction of a cooperative bimetallic catalytic system with two collaborative metal sites for three-component Strecker reactions.

[1] Department of Chemistry, Texas A&M University, College Station, TX 77843-3255, USA. [2] Department of Materials Science and Engineering, Texas A&M University, College Station, TX 77843-3003, USA. [3] School of Chemistry and Chemical Engineering, Shandong University, 250100 Jinan, P. R. China. [4] Department of Chemistry, College of Science, King Saud University, Riyadh, 11451, Saudi Arabia. [5] Artie McFerrin Department of Chemical Engineering, Texas A&M University, College Station, TX 77843-3022, USA. Shuai Yuan and Jun-Sheng Qin contributed equally to this work. Correspondence and requests for materials should be addressed to H.-C.Z. (email: zhou@chem.tamu.edu)

As an inorganic and organic hybrid material, metal–organic frameworks (MOFs) epitomize ideal tunable platforms by judicious design of metal nodes and organic linkers[1–4]. Indeed, the ability to control MOF structures and structure-related properties at the molecular level has prompted numerous advances in basic sciences, especially in supramolecular chemistry[5–11]. Furthermore, this new level of control has led to fascinating developments with respect to applications in gas storage, separation, drug delivery, and catalysis[12–16]. Quests for advanced functionalities in MOFs typically require more complex structures and pore environments, accompanied by heightened challenges concerning their geometric design[17]. Constructing MOFs from multiple components is a pathway to achieve highly complex crystalline materials for sophisticated applications. In a recent review article, Yaghi and co-workers proposed that the future MOFs should have multiple building units, the arrangement of which should have specific sequences within crystals[18]. MOFs built from multiple constituents have revealed emerging properties that are beyond the linear integration of those from the single components[19]. Therefore, research on multi-component MOFs has accelerated rapidly in recent years, and has expanded the complexity and diversity of known porous materials[20–25].

A simple approach to incorporate multiple components into a MOF is to employ a set of organic linkers of the same length, geometry, and connectivity, yet with different substituents. Yaghi and co-workers have incorporated as many as eight different functionalities into a MOF-5 backbone that produced multivariate MOFs[19]. However, this method typically lacks a high level of control over the position of functional groups[26]. To this end, another strategy is to use multiple linkers with different symmetry or connectivity that can be differentiated in the crystal lattice[27]. This strategy has been well demonstrated by Telfer and co-workers, who reported the synthesis of a quaternary MOF with three topologically distinct carboxylate linkers[21,22]. Given the complexity of the quaternary system, it is exceedingly challenging to predict the resulting structure and avoid the multiple phases that compete for MOF formation. To address those challenges, we recently demonstrated that sequentially installing linear linkers in coordinatively unsaturated MOFs by post-synthetic modification can predictably yield a quaternary MOF[28].

In addition to the mixed-linker strategy, the complexity of MOFs can also be increased by employing multiple metal clusters or inorganic secondary building units (SBUs)[29–31]. Li and co-workers reported a one-pot synthesis of a quaternary MOF combining Cu-based triangular, Zn-based octahedral, and Zn-based square pyramidal SBUs with only one organic linker[31]. Compared with mixed-linker MOFs, it is more challenging to synthesize multi-component MOFs with different inorganic SBUs. The main reason is the sensitivity of cluster formation to reaction conditions and, in many cases, the incompatibility of the formation conditions of different clusters[32]. This limitation has prevented the integration of different inorganic SBUs into MOFs and the potential properties they could provide.

Since more sophisticated architectures are needed as the applications of MOFs continue to expand, the growing gap between design and synthesis has become a critical limitation. The complexity of multi-component MOFs is limited ultimately by the lack of control during the framework assembly in a one-pot reaction. Indeed, the competitive assembly of multiple components usually ends up with a thermodynamically favored MOF as the main product, which impedes the incorporation of multi-components into a kinetic product. Retrosynthetic analysis is a concept to approach the synthesis of complex organic molecules by transforming a target molecule into simpler precursors and sequentially applying a set of known chemical reactions to assemble them together. Inspired by the concept of retrosynthesis utilized to construct complex organic molecules and natural products[33], we proposed that MOFs as porous solid materials could be rationally designed and synthesized in multi-step synthetic approaches[34]. Therein, kinetic analysis, an analog of retrosynthetic analysis, allowed us to conceptually break down a predicted MOF structure into available metal precursors and ligands (i.e., synthons). The synthons were subsequently assembled into the designed MOF structure step by step under kinetic control using known post-synthetic modification methods[34,35], which circumvent the undesirable thermodynamic sink. In other words, we use labile coordination bonds to sequentially layer-on molecular elaborations to a robust framework, which eventually formed the designed multi-component coordination assemblies. As a proof of concept, four quaternary MOFs were assembled under the kinetic guidance, each composed of up to three different metals and two different linkers in a predetermined array within the crystal lattice. Furthermore, to demonstrate the utility of such a retrosynthetic design, a cooperative bimetallic catalytic system was constructed in a MOF by the sequential incorporation of a Fe-porphyrin and a Cu-pyridyl moiety.

## Results

**Kinetic analysis.** Chemists approach the synthesis of complex organic molecules by identifying simpler molecules that can be modified rationally or linked through covalent bonds by sequentially applying a set of known chemical reactions (Fig. 1a)[34]. Organic synthesis takes robust covalent bond as its base, which

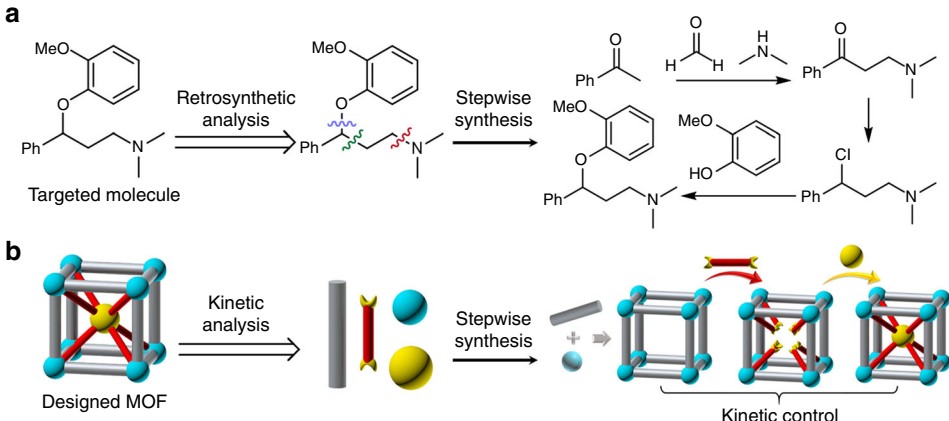

**Fig. 1** Schematic representations. Retrosynthesis of organic molecules (**a**) and multi-component MOFs (**b**)

allows for further modification of functional groups or connection with other moieties without breaking the molecular backbone. Researchers have thus developed and applied numerous chemical reactions that together form a reaction library. The idea of MOF retrosynthesis conceptually parallels the retrosynthetic analysis used for organic synthesis: organic linkers and metal clusters are sequentially linked by coordination bonds to form pre-designed MOF structures (Fig. 1b). However, fundamental differences between organic molecules and MOFs might hamper the realization of MOF retrosynthesis. Unlike covalently linked organic molecules, MOFs are mostly composed of relatively labile coordination bonds, which could break when multiple modification reactions are performed post-synthetically. To address this, we proposed a kinetic analysis for the design and construction of multi-component MOFs. First, the proposed MOF structure is broken down into available organic linkers and metal clusters. Second, the organic linkers and metal clusters as synthons are sequentially assembled under kinetic guidance. In order to maintain the structural integrity of the MOF during multi-step reactions, the most robust coordination bond, which usually requires harsh solvothermal reaction conditions, is formed initially. Other synthons are subsequently anchored onto the framework by labile coordination bonds formed under relatively mild condition. The whole process is controlled by kinetics so that different components can be precisely placed at designated positions within the crystal lattice while maintaining the structural intactness of the MOF backbone to the maximum extent. Therefore, multi-component MOFs as kinetic products could be isolated by circumventing the undesirable thermodynamic sink in a one-pot reaction. Since previous studies have demonstrated the feasibility of post-synthetic incorporation of two essential parts of MOFs, organic linkers, and inorganic metal clusters[28,35], we have reason to believe that multi-component MOFs can be synthesized by sequential installation of linkers and metal clusters in a prototype MOF through a series of known chemical reactions.

We initially sought for a parent MOF that could be used as a scaffold for the incorporation of organic and inorganic components. To be used for such purposes, three criteria need to be satisfied. First, there must exist binding sites for the installation of hetero-linkers or clusters. Second, the cavities should be large enough to provide sufficient space for the subsequent incorporation of metal–organic moieties. Third, the framework needs to be stable to ensure the structural intactness during multiple modification reactions. Following an analysis of reported MOFs, we found that a zirconium-MOF, PCN-224, fulfilled all three criteria[36]. First, PCN-224 is constructed from 6-connected $Zr_6$ clusters, leaving six pairs of terminal $H_2O/OH^-$ groups poised for the binding of carboxylates. Studies have shown that $Zr_6$ clusters with unsaturated connectivity readily coordinate with carboxylate ligands by replacing the terminal $H_2O/OH^-$ groups via acid and base reactions[37,38]. Second, PCN-224 is highly porous with a large cavity (~2 nm in diameter), which is sufficient for further installation of organic linkers and incorporation of inorganic clusters. Third, the high chemical stability of PCN-224 ensures its structure intactness during harsh modification conditions. Therefore, PCN-224 is selected as an ideal candidate for the incorporation of secondary linkers and metal clusters.

The PCN-224 structure contains a large cubic cage confined by eight $Zr_6$ clusters (Fig. 2a). Three pairs of terminal $H_2O/OH^-$ from each $Zr_6$ cluster are exposed to the inner wall of the cage (Fig. 2b). Noticeably, terminal $H_2O/OH^-$ groups from each pair of neighboring $Zr_6$ clusters can be connected by bent ditopic fragments (Fig. 2d). Thus, a metal-isonicotinate (M-INA) fits the space ideally. As shown in Fig. 2c, a model, namely PCN-201, was built by installing M-INA between each pair of adjacent $Zr_6$ clusters. A careful examination of the PCN-224 structure

indicates that the terminal $H_2O/OH^-$ groups from each pair of neighboring $Zr_6$ clusters can be bridged by tritopic linkers, thereby forming a cube in which each linker acts as an edge (Fig. 2f). The cube contains 12 edges, each with a dangling carboxylate pointing toward the cubic center, so that a 12-connected inorganic SBU will fit into the center. Considering the lengths of linkers and size of clusters, we proposed that the combination of a tritopic organic linker and a 12-connected inorganic SBU would fit the cubic cage of PCN-224, thus forming a new MOF named PCN-202 (Fig. 2e).

**Retrosynthesis of multi-component MOFs.** Guided by kinetic considerations, the robust PCN-224 was initially synthesized by the reaction of $ZrCl_4$ and TCPP (TCPP = tetrakis(4-carboxyphenyl)porphyrin) at 120 °C with benzoic acid as the modulating reagent (Fig. 3a, b). The crystals of PCN-224 were washed with water and 1 M HCl/DMF (DMF = N,N-dimethylformamide) solution to remove dangling ligands on the $Zr_6$ clusters and the weakly bonded $Zr^{4+}$ on the porphyrin center. The vacant porphyrin center readily chelated transition metals with +2 or +3 oxidation states. Because of the chelating effect, the metalloporphyrin became robust enough to survive further modification reactions. As a proof of concept, the porphyrin center of PCN-224 was metallated with $Ni^{2+}$ by incubating PCN-224 crystals in $Ni(NO_3)_2$/DMF solution (70 mM) at 100 °C overnight (Fig. 3c). The single-crystal structure clearly indicated the existence of $Ni^{2+}$ at the center of the porphyrin linker. The crystals of PCN-224(Ni) were further treated with the INA solution in DMF (30 mM) resulting in the selective binding of INA to the coordinatively unsaturated sites of $Zr_6$ clusters through carboxylates (Fig. 3d). According to the hard and soft acids and bases theory, the carboxylate group will bind to the coordinatively unsaturated sites of $Zr_6$ clusters by replacing the terminal $H_2O/OH^-$ groups leaving the pyridyl moiety open. A pair of pyridyl groups from neighboring $Zr_6$ clusters forms a vacant bipyridyl moiety allowing the binding of most transition metal ions under proper conditions. As a proof of concept, a $Cu^+$ or a $Ni^{2+}$ ion was installed into the PCN-224(Ni)-INA to form PCN-201(Ni)-Cu and PCN-201(Ni)-Ni, respectively. The $Cu^+$ ions were installed in PCN-224(Ni)-INA by reacting with CuI solution in MeCN (70 mM) at 65 °C (Fig. 3f). Similarly, the bipyridyl moiety of PCN-224(Ni)-INA was metallated with $Ni^{2+}$ by treating with $NiCl_2$ solution in DMF (70 mM) at 65 °C (Fig. 3g). PCN-201(Ni)-Cu and PCN-201(Ni)-Ni were characterized by single-crystal X-ray diffraction (SC-XRD), which clearly showed the existence and coordination environment of the subsequently incorporated $Cu^+$ and $Ni^{2+}$. According to the crystal structure, the $Cu^+$ center was chelated by two pyridyl groups from INA ligands and further coordinated with a solvent and an $I^-$ counterion, thereby achieving a tetrahedral geometry. On the other hand, $Ni^{2+}$ formed a square planar coordination environment by coordinating with two pyridyl groups from INA ligands and two $Cl^-$ as counterions.

Similar experiments were carried out to synthesize the proposed structure of PCN-202 using tritopic linkers, TPA (4,4′,4″-tricarboxytriphenylamine), for example, but to no avail. We reason that the formation of the proposed structure required TPA in an unfavorable conformation. In order to fit the pore environment of PCN-224, TPA needs to adopt a $C_{2v}$ symmetry with each phenyl ring perpendicular to the equatorial plane as shown in Fig. 4a. However, TPA molecules tend to exhibit a $C_3$ symmetry (Fig. 4b), which is not compatible with the PCN-224 structure. To solve this problem, we employed DCDPS (4,4′-dicarboxydiphenyl sulfone) as a linker instead of TPA. Although DCDPS is a ditopic linker with a bent conformation (Fig. 4c), it crystallographically appears as a triangular linker if it is

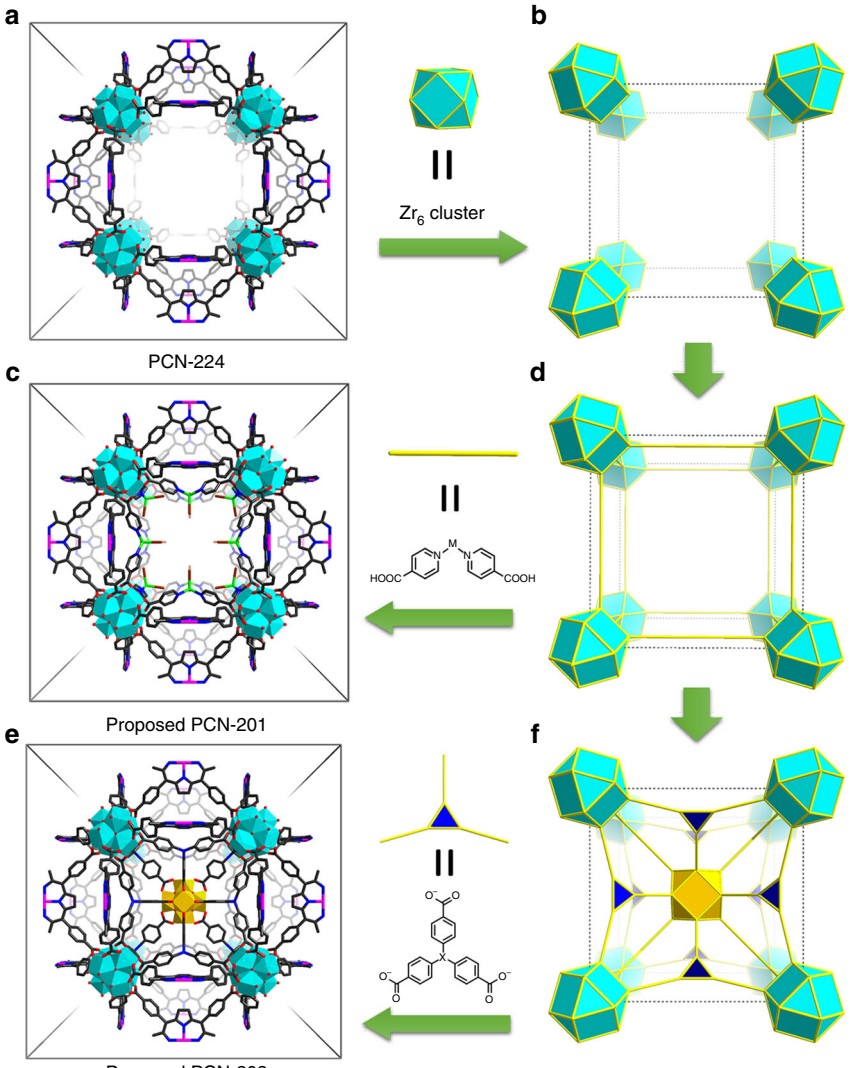

**Fig. 2** Design of multi-component MOFs based on PCN-224 prototype. **a** The unit cell of PCN-224 and **b** the cubic cage confined by eight clusters. **c** The unit cell of proposed structure PCN-201 and **d** the topological representation of PCN-201 with linear metal-INA fragments incorporated. **e** The unit cell of proposed structure PCN-202 and **f** the topological representation of PCN-202 with a tritopic linker and 12-connected cluster incorporated

disordered (Fig. 4d). Each phenyl ring of DCDPS was perpendicular to the equatorial plane, which ideally matched the pore environment of PCN-224. Compared to TPA, DCDPS is a semi-flexible linker that can adjust its conformation to a certain extent. This will ensure the successful coordination of DCDPS to the cavity of PCN-224 even if the linker length and bond angle do not exactly match the pore environment.

Crystals of PCN-224(Ni) were treated with a DMF solution of DCDPS (30 mM, 15 mL) at 75 °C for 24 h (Fig. 3e). As shown by the crystal structure, DCDPS was coordinated on the vacant sites of $Zr_6$ clusters by replacing terminal $H_2O/OH^-$ groups. Given the flexible conformation of the dangling DCDPS and the disorder caused by the framework symmetry, it is difficult to precisely determine the occupancy of DCDPS in this intermediate structure by crystallography. Based on the proton nuclear magnetic resonance ([1]H-NMR) data of the digested samples, the DCDPS: TCPP ratio is about 4:1, which equals to 6 DCDPS on each $Zr_6$ clusters. This result indicates that all terminal $H_2O/OH^-$ groups are replaced by a dangling DCDPS with only one coordinated carboxylate group. After the DCDPS treatment, a solution of $HfCl_4$ (100 mM) and AcOH (400 mM) in DMF (5 mL) was added. The mixture was heated at 80 °C for 24 h to afford crystals

of PCN-202(Ni)-Hf (Fig. 3h). PCN-202(Ni)-Zr was also assembled under similar synthetic conditions except that $ZrCl_4$ was used instead of $HfCl_4$ (Fig. 3i). The crystal structure of PCN-202(Ni)-Hf and PCN-202(Ni)-Zr showed the same backbone structure as PCN-224(Ni) with $Hf_6$ (or $Zr_6$) clusters in the pore center and DCDPS bridging the neighboring $Hf_6$ (or $Zr_6$) clusters. It should be noted that the conformation of the linkers was important for the formation of PCN-202. Two other linkers, OBC and CDC, with $C_2$ and $C_{2v}$ symmetry, respectively, have also been attempted to construct PCN-202 structures (OBC = 4,4′-oxybis-benzoate, CDC = 3,6-carbazoledicarboxylate). Although they have approximately the same size with DCDPS, the dihedral angles between the carboxylate group and the equatorial plane (~40° for OBC and 0° for CDC) are not suitable for the PCN-202 structure. Therefore, they did not give rise to PCN-202 under identical synthetic condition.

The SC-XRD data unambiguously showed the stepwise incorporation of organic linkers and formation of metal clusters, which ruled out the possibility of MOF dissolution and recrystallization. Furthermore, microscopic observation of the crystal during modification showed no change in crystal size or shape (Supplementary Figures 1 and 2). Moreover, the solution

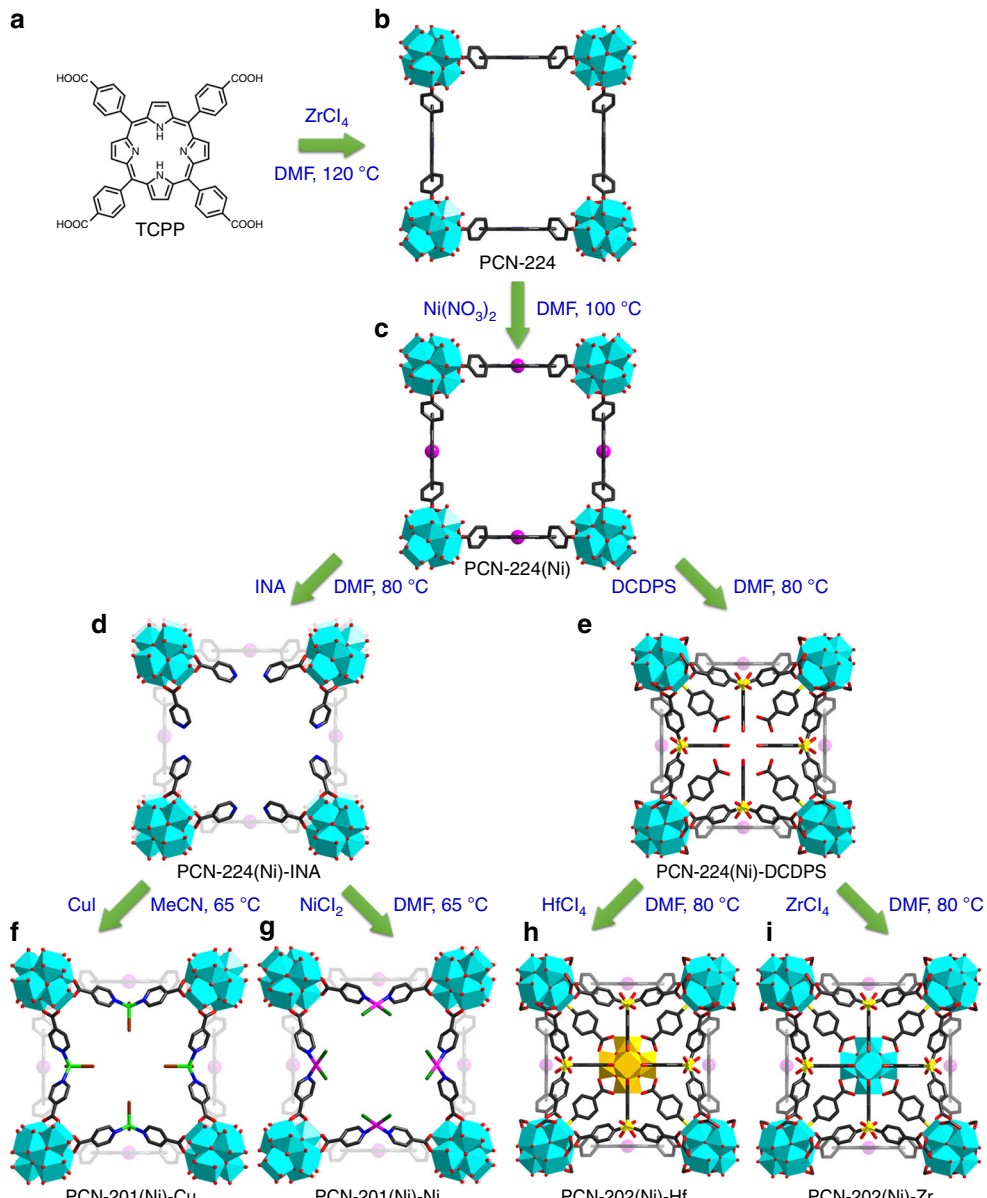

**Fig. 3** Synthetic approaches to multi-component MOFs. **a** TCPP linker, **b** PCN-224, **c** PCN-224(Ni), **d** PCN-224(Ni)-INA, **e** PCN-224(Ni)-DCDPS, **f** PCN-201(Ni)-Cu, **g** PCN-201(Ni)-Ni, **h** PCN-202(Ni)-Hf, and **i** PCN-202(Ni)-Zr

was colorless throughout the treatment with DCDPS and HfCl$_4$ solution, thereby indicating no dissolution of PCN-224(Ni) that would release dark red TCPP. The supernatant was further separated and analyzed by ultraviolet (UV) and inductively coupled plasma mass spectrometry (ICP-MS), which eliminated the possible existence of TCPP or Zr$^{4+}$ that could be generated by the decomposition of PCN-224(Ni) (Supplementary Figure 3). Such results confirmed that the formation of PCN-201 and PCN-202 from PCN-224(Ni) was a single crystal to single-crystal transformation process instead of a dissolution-recrystallization process.

The formation process of PCN-201 and PCN-202 was carried out under rigorous kinetic control. The assembly of all the components in a one-pot reaction always ends up with thermodynamically favored PCN-224 as the main product, whereas other labile components were excluded from the product. The sequence of each process was also designed by kinetic considerations, which were essential for the formation of the targeted structure. For example, if the porphyrin center was initially unoccupied, Cu-porphyrin would form during the treatment with CuI solution, which could hardly be replaced by other metals given the strong chelating effect of porphyrin. The sequence of linker installation and cluster incorporation was also critical for the final structure. When the CuI and INA were simultaneously introduced to the reaction system, Cu-INA-based coordination polymers were immediately formed as impurities. Likewise, if HfCl$_4$ and DCDPS were added simultaneously, those two components tend to form a gel in the solution instead of an ordered structure in the MOF cavity. Furthermore, the one-pot synthesis of PCN-201 and PCN-202 were attempted starting from a mixture of organic linkers and metal salts, which did not give rise to desired products. Indeed, the self-assembly of organic linkers and metal cations in a one-pot reaction is essentially a black box. Different combinations of metal cations and ligands make it exceedingly difficult to control the product. More importantly, the formation condition of different metal–organic species in a multi-component MOF requires different solvents, temperatures, and metal-ligand ratios, which are almost

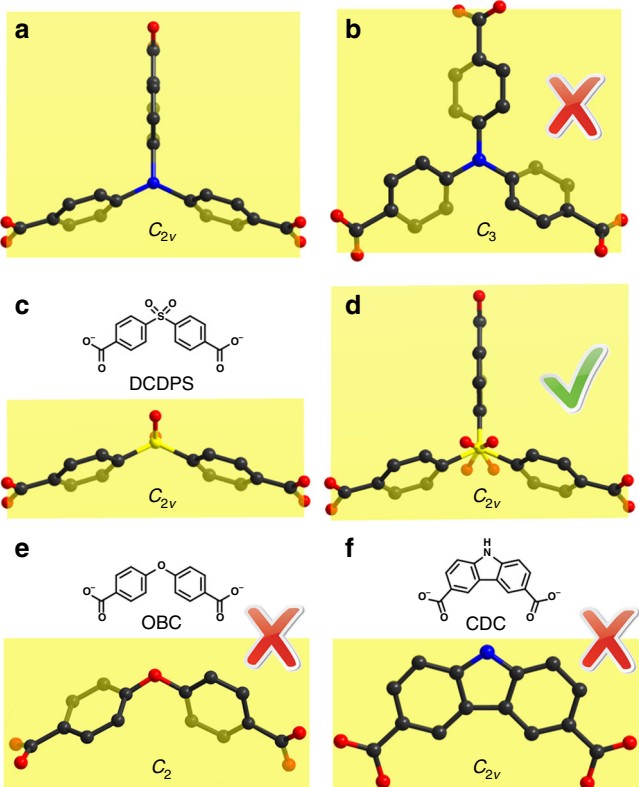

**Fig. 4** Geometries of ligands. **a** TPA conformation in the proposed structure, **b** TPA conformation in its free state, **c** DCDPS conformation in its free state, **d** disordered DCDPS linker, **e** and **f** conformation and symmetry of OBC and CDC

impossible to satisfy in a one-pot reaction. Indeed, MeCN as solvent favors the dissolution of CuI and the formation of CuI-based MOFs but disfavors the crystallization of Zr-based MOFs. Meanwhile, the synthesis of Zr-based MOFs usually requires an excess amount of acid as modulating reagent, which prohibits the formation of CuI or Ni-based MOFs. The multi-step synthesis with kinetic considerations allows the integration of different metal SBUs into one framework by avoiding the undesired thermodynamic sink. The successful synthesis of PCN-201 and PCN-202 highlights the power of kinetic analysis for the construction of multi-component MOFs.

**Structure description.** PCN-201(Ni)-Cu and PCN-202(Ni)-Hf represent rare cases of quaternary MOFs composed of up to three different metals and two different linkers, each compartmentalized in a predetermined array within the crystal lattice. For clarity, the structures of PCN-201(Ni)-Cu and PCN-202(Ni)-Hf were described and compared with their parent PCN-224(Ni). PCN-201(Ni)-Cu crystallized in the cubic space group $Im$-$3m$ (Supplementary Table 1). Crystallographically, it contains a 12-connected $Zr_6$ cluster, a pair of $Cu^+$ bridged INA (Cu-INA$_2$), and a metalloporphyrin linker with $Ni^{2+}$ in the center. Each $Zr_6$ cluster was connected by six adjacent TCPP linkers forming the scaffold structure (PCN-224), which can be simplified into a 4,6-connected net with **she** topology (Fig. 5a, e). Each Cu-INA$_2$ moiety bridges the neighboring $Zr_6$ cluster to fulfill the structure of PCN-201(Ni)-Cu. Topologically, the 12-connected metal clusters can be regarded as cuboctahedron nodes and tetratopic TCPP linkers can be viewed as square nodes (Fig. 5d). The Cu-INA$_2$ moiety was treated as a linear linker for clarity. The overall

structure was analyzed to be a 4,12-connected net with a point symbol of $\{3^{12}.4^{18}.5^{24}.6^{12}\}_2\{3^4.4^2\}_3$ determined by TOPOS 4.0 (Fig. 5b, f)[39]. The new topology was named as **tam** and included in the reticular chemistry structure resource (RCSR) database[40].

Single-crystal X-ray analysis revealed that PCN-202(Ni)-Hf also crystallized in the cubic space group $Im$-$3m$ (Fig. 5c). It contains a 12-connected $Zr_6$ cluster, a 12-connected $Hf_6$ cluster, a disordered DCDPS linker, and a $Ni^{2+}$ centered metalloporphyrin linker. Each component is precisely located at a predetermined position within the periodic lattice. It has the same backbone structure as PCN-224(Ni), which is formed by the connection of 6-connected $Zr_6$ clusters and tetratopic TCPP linkers. The DCDPS is twofold disordered in the crystal lattice so that it appears as a 3-connected linker. Crystallographically, 12 tritopic DCDPS linkers bridged eight neighboring $Zr_6$ clusters into a cube with side lengths of ~2 nm. Each tritopic DCDPS linker on the edge of the cube is connected to a pair of $Zr_6$ clusters, thereby leaving a dangling carboxylate pointing toward the cube center. The 12 dangling carboxylates form a natural pocket for a 12-connected cluster, which ideally accommodates a $Hf_6$ cluster. Topologically, the 12-connected metal cluster can be regarded as cuboctahedron nodes and tetratopic TCPP linkers can be viewed as square nodes (Fig. 5d). The DCDPS linker is twofold disordered in the structure so that it is regarded as a 3-connected triangle node for simplicity. The overall structure can be simplified into a 3,4,12,12-connected net with a point symbol of $\{4^{24}.6^{36}.8^6\}_5\{4^3\}_{12}\{4^4.6^2\}_6$, namely **amu** (Fig. 5g, Supplementary Figure 4a and d).

To clarify the connection of the disordered DCDPS linker, we simulated an ordered structure of PCN-202(Ni)-Hf by reducing the space group from $Im$-$3m$ to $I$-$43m$ (Supplementary Figure 4a and b). The reduced space group will eliminate the disorder of the DCDPS linker by removing the mirror plane passing through its center. In the simulated structure, each DCDPS linker is 2-connected to a $Zr_6$ cluster and a $Hf_6$ cluster. Therefore, each $Zr_6$ cluster is 9-connected to six TCPP and three DCDPS, respectively, while each $Hf_6$ cluster is 12-connected to DCDPS linkers. Three pairs of DCDPS linkers bridge a pair of $Zr_6$ and $Hf_6$ clusters so that they are topologically simplified into an edge (Supplementary Figure 5). Consequently, $Zr_6$ clusters are regarded as 5-connected hexagonal pyramid nodes while $Hf_6$ clusters are reduced into 4-connected tetrahedron nodes (Supplementary Figure 4c). The overall structure is simplified into a 4,4,7-connected net with a point symbol of $\{4^4.6^2\}_6\{4^6.6^{15}\}_4\{6^6\}$, namely **hcz** (Supplementary Figure 4e). Note that the topology of ordered structure depends on the space group that is chosen to eliminate the disorder. Different topologies might have resulted if other space groups were selected to simplify the structure.

**Stability and porosity.** Based on the single-crystal structure, $^1$H-NMR and elemental analysis data, the formula of PCN-201 (Ni)-Cu, PCN-201(Ni)-Ni, PCN-202(Ni)-Hf, and PCN-202(Ni)-Zr were determined as $(Zr_6O_4(OH)_4)_4(CuI)_{12}TCPP_6INA_{24}$, $(Zr_6O_4(OH)_4)_4(NiCl_2)_{12}TCPP_6INA_{24}$, $(Zr_6O_4(OH)_4)_4(Hf_6O_4(OH)_4)TCPP_6DCDPS_{12}HDCDPS_6(OH)_6(H_2O)_6$, and $(Zr_6O_4(OH)_4)_5TCPP_6DCDPS_{12}HDCDPS_6(OH)_6(H_2O)_6$, respectively (Supplementary Figures 6–9). Their compositions were further confirmed by energy-dispersive X-ray spectroscopy (EDX), ICP-MS experiments (Supplementary Tables 2 and 3). Elemental mapping at the microstructural level by scanning electron microscope (SEM) with EDX showed a uniform distribution of metals and ligands throughout the crystal, indicating a homogeneous crystalline material (Supplementary Figure 29). The incorporation of hetero-metal-clusters and organic linkers slightly

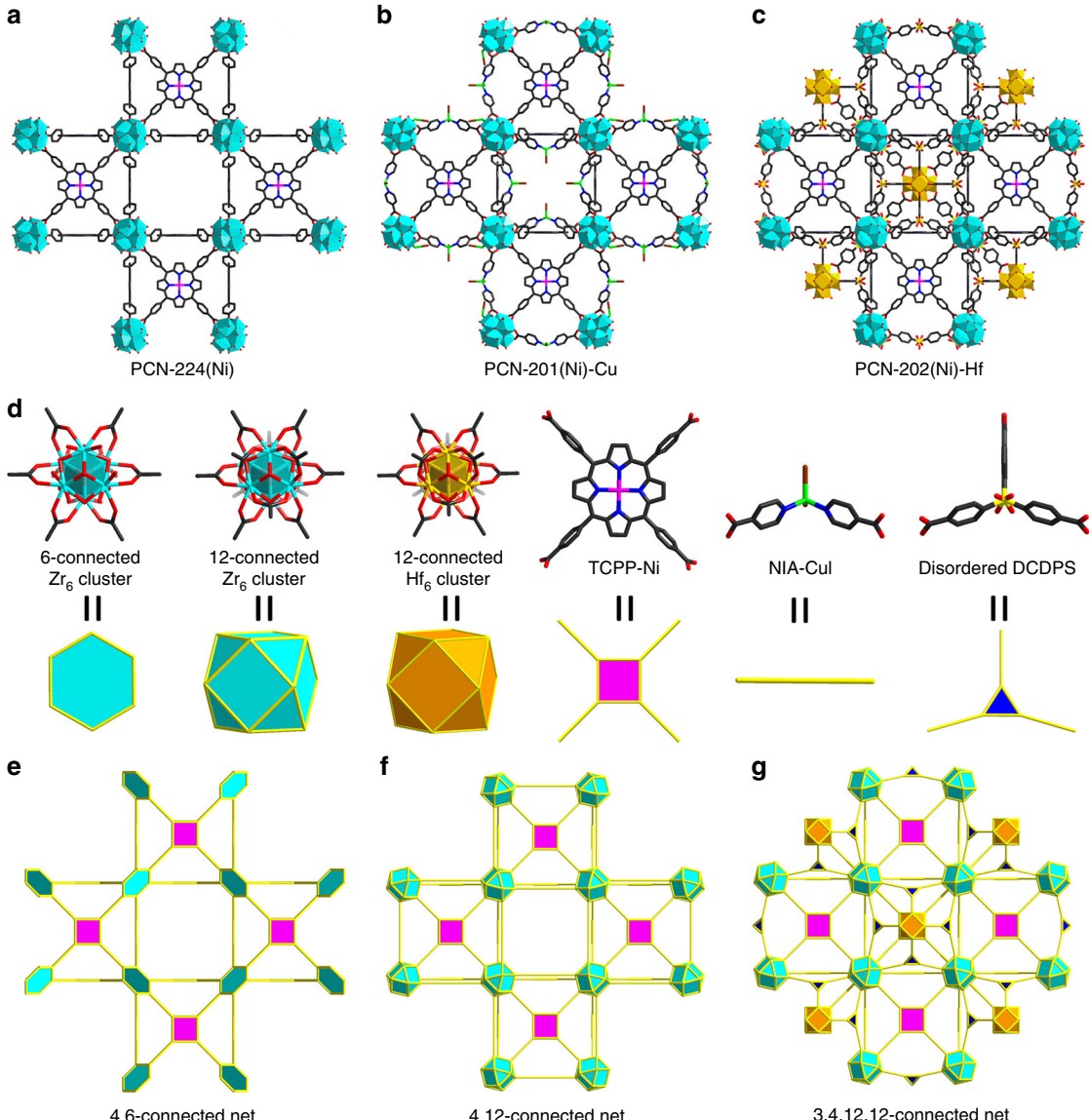

**Fig. 5** Comparison of structures. **a**, **b**, **c** Crystal structures of PCN-224(Ni), PCN-201(Ni)-Cu, and PCN-202(Ni)-Hf, **d** building units and their simplified topological elements, **e**, **f**, **g** topologies of PCN-224(Ni), PCN-201(Ni)-Cu, and PCN-202(Ni)-Hf

changed the X-ray diffraction of the material as recognized on powder X-ray diffraction (PXRD) patterns (Supplementary Figures 10–13). New peaks emerge at 7.2, 10.7, and 11.7° (2θ) in the PXRD pattern of PCN-201(Ni)-Cu, which matched well with simulations. In the PXRD of PCN-201(Ni)-Ni, the peak at 3.3° disappears, while two additional peaks appear at 7.3 and 10.9°. The PXRD patterns of PCN-202(Ni)-Zr and PCN-202(Ni)-Hf are almost identical and differ from PCN-224(Ni) at 8.6, 10.8, and 11.8°.

PCN-224(Ni), PCN-201(Ni)-Cu, PCN-201(Ni)-Ni, PCN-202 (Ni)-Hf, and PCN-202(Ni)-Zr were also analyzed by thermo-gravimetric analysis (TGA) (Supplementary Figures 14 and 15). The initial weight loss before 185 °C is attributed to the removal of the water molecules in the pores, whereas the removal of coordinated water on the $Zr_6$ cluster corresponds to the weight loss from 185 to 265 °C (Supplementary Figure 16). PCN-224(Ni) shows much higher weight loss than other samples before 265 °C due to the water removal from $Zr_6$ clusters (7.576% for PCN-224 (Ni) and ~1% for other samples). Presumably, the terminal $H_2O$/ $OH^-$ groups on the $Zr_6$ clusters are replaced by carboxylate

groups in PCN-201(Ni)-Cu, PCN-201(Ni)-Ni, PCN-202(Ni)-Hf, and PCN-202(Ni)-Zr so that the water content in PCN-224(Ni) is far greater than other MOFs. The decomposition of PCN-201 (Ni)-Cu starts at 300 °C, whereas other MOFs do not show obvious mass loss until 400 °C. This is tentatively attributed to the redox active Cu that catalyzes the oxidative decomposition of the framework. The further weight losses corresponding to the thermal decomposition of organic fragments match well with the calculation.

The $N_2$ adsorption isotherms of PCN-201(Ni)-Cu, PCN-201 (Ni)-Ni, PCN-202(Ni)-Hf, and PCN-202(Ni)-Zr were measured and show a clear decrease of $N_2$ total uptake compared with PCN-224(Ni), corresponding to the cavity filled by subsequently introduced linkers and clusters (Fig. 6a and Supplementary Figure 17). More importantly, the large cavity observed in the crystal structure of PCN-224(Ni) with a diameter of ~2 nm was occupied by metal–organic species so that a decrease in the pore diameter was observed by the pore size distributions derived from $N_2$ isotherms (Fig. 6b, Supplementary Figures 18–20). The subsequently incorporated component significantly changed the

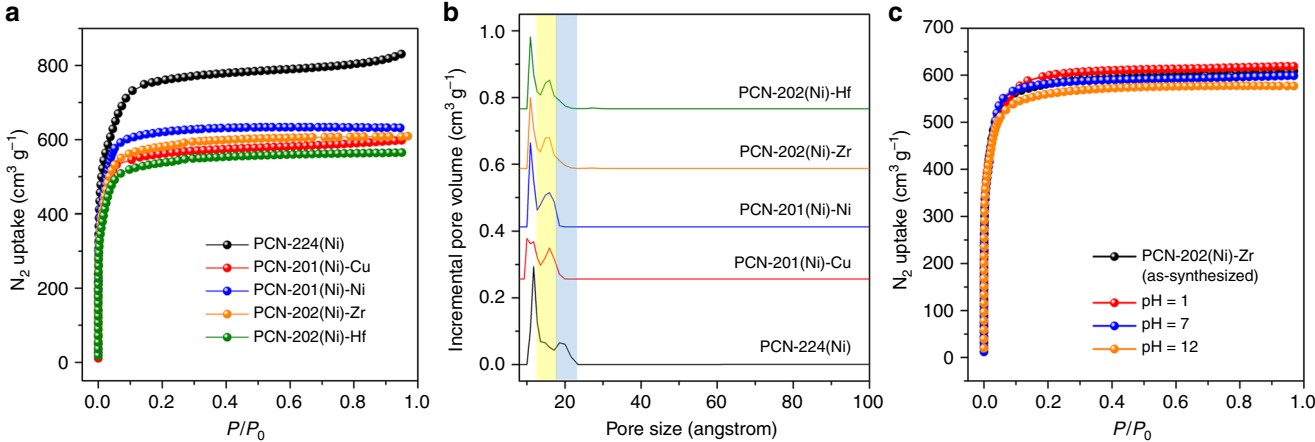

**Fig. 6** Gas sorption measurements for PCN-224(Ni) and its derivatives. **a** $N_2$ adsorption isotherms of PCN-224(Ni) and its derivatives at 77 K. **b** Pore size distributions of PCN-224(Ni) and its derivatives calculated from $N_2$ adsorption isotherms. **c** $N_2$ adsorption isotherms of PCN-202(Ni)-Zr upon treatments in pH = 1, 7, and 12 aqueous solutions

pore environment, which in turn affected the gas adsorption properties. Indeed, the volumetric $H_2$ adsorption capacity of PCN-202(Ni)-Zr increased by 64% compared to the parent framework PCN-224(Ni) (Supplementary Figures 21 and 22).

The stabilities of PCN-201(Ni)-Cu, PCN-201(Ni)-Ni, PCN-202 (Ni)-Hf, and PCN-202(Ni)-Zr were also tested. By virtue of the robust Zr (or Hf)-carboxylate bond, PCN-202(Ni)-Hf and PCN-202(Ni)-Zr show high stability in aqueous solution with pH ranging from 1 to 12 (Fig. 6c). The PXRD patterns and $N_2$ isotherms verified the intactness of frameworks after the stability test (Supplementary Figures 23–25). [1]H-NMR and ICP results of digested MOF samples further confirmed that the composition of materials remained unaffected by the acid or base treatment. Interestingly, PCN-201(Ni)-Cu and PCN-201(Ni)-Ni also show good water stability although the labile Cu-pyridyl or Ni-pyridyl are often believed to be water sensitive (Supplementary Figure 26). This phenomenon indicated that the robust MOF backbone could reinforce the labile moieties in the structure, which has been previously observed in multi-component MOFs composed of different inorganic SBUs[29]. It should be noted that the growth of a secondary metal node on the installed linker stabilized the dangling linkers by reinforcing them. For example, the dangling INA or DCDPS ligands on PCN-224(Ni)-INA and PCN-224(Ni)-DCDPS can be easily removed by water treatment. However, once the ligand bound with metals to form PCN-201(Ni)-Ni and PCN-202(Ni)-Hf, the entire structure became water stable. The reinforcement effect suggests the possibility of incorporating catalytically active metal clusters into a robust MOF by stepwise synthesis, which is otherwise difficult to realize since some active metal clusters are often too labile to support a stable framework.

**Cooperative bimetallic catalysis**. The retrosynthesis of multi-component MOFs affords a new level of control over MOF structures and their related properties, which leads to highly tunable multifunctional MOF systems. Herein, we demonstrate that a cooperative bimetallic catalytic system can be constructed in PCN-201(Fe)-Cu by introducing Fe-porphyrin and Cu-pyridyl simultaneously. The Fe-porphyrin and Cu-pyridyl moieties cooperate as a Lewis acid and a nucleophile activator, respectively, leading to improved catalytic activity toward three-component, one-pot Strecker reactions.

Cooperative multimetallic activation is a common feature in enzyme catalysis[41]. In many enzymes, two or more metal centers in the active site are able to activate both nucleophilic and electrophilic reactants leading to improved kinetics and higher selectivity. Inspired by these natural systems, researchers are now developing dual activation catalysts[42]. For example, the combination of Lewis acids and Lewis bases in a single catalytic system has been applied in cycloaddition reactions between ketene enolates and various electrophiles, cyanosilylation of ketones, Corey–Chaykovsky epoxidation, and catalytic Strecker-type reactions[43,44]. The Strecker reaction is a versatile way of preparing α-aminonitriles through the attack of a nitrile group to an imine group[45]. The resulting α-aminonitriles can be hydrolyzed to obtain α-amino acids or used as intermediates in the preparation of nitrogen-containing heterocycles[46,47]. The three-component, one-pot Strecker reaction goes through in situ imine formation and cyano group addition, which relies on the activation of both nucleophile and electrophile[48]. When mono-nuclear catalysts are used, such double activation is revealed by second-order reaction kinetics illustrating that two catalyst nuclei are involved in the transition state. Therefore, catalysts containing multiple metal centers in an appropriate proximity and arrangement can result in cooperative reaction pathway and better reactivity than the related monometallic systems[49]. In this context, multi-component MOFs represent a convenient path to incorporate multiple catalytic sites in a predetermined array defined by the crystal lattice, which will lead to promising cooperative catalytic effects in a solid state material[50,51]. Coupling two catalytic sites in a MOF offers several additional advantages over homogeneous systems, including easy catalyst separation and recovery, regeneration, and handling.

PCN-201(Fe)-Cu as a cooperative bimetallic catalytic system was assembled via stepwise linker installation and metalation. First, PCN-224(Fe) was synthesized by the metalation of PCN-224 with FeCl$_2$ at 100 °C. The Cu-INA fragments were subsequently assembled in PCN-224(Fe) to generate PCN-201 (Fe)-Cu. PCN-201(Fe)-Cu is iso-structural to PCN-201(Ni)-Cu except for the $Fe^{3+}$ on the porphyrin center. The catalytic performance of the PCN-201(Fe)-Cu system in three-component, one-pot Strecker reactions is evaluated using benzaldehyde, aniline, and TMSCN (trimethylsilyl cyanide). A catalytic amount of PCN-201(Fe)-Cu leads to the quantitative formation of the α-aminonitrile with a turnover frequency (TOF) of 6000 per h. For comparison, the catalytic activities of two reference MOF catalysts, as PCN-201(Ni)-Cu and PCN-224(Fe) have also been examined under similar experimental conditions. PCN-201(Ni)-Cu contains the Cu-pyridyl sites, whereas the Lewis acidic Fe-porphyrin center was replaced by inert Ni-porphyrin. PCN-224

**Table 1 Catalytic performance of PCN-201(Fe)-Cu in the three-component Strecker reaction[a]**

| Entry | Substrate | Catalyst | Yield (%) |
|---|---|---|---|
| 1 | Benzyl aldehyde | PCN-201(Fe)-Cu | 99 |
| 2 | Benzyl aldehyde | PCN-201(Ni)-Cu | 21 |
| 3 | Benzyl aldehyde | PCN-224(Fe) | 24 |
| 4 | Benzyl aldehyde | PCN-201(Ni)-Cu and PCN-224 (Fe) mixture | 68 |
| 5[b] | 4-Methylbenzyl aldehyde | PCN-201(Fe)-Cu | 99 |
| 6 | 4-Methoxybenzaldehyde | PCN-201(Fe)-Cu | 92 |
| 7[b] | 4-Chlorobenzyl aldehyde | PCN-201(Fe)-Cu | 96 |
| 8[b] | 4-Bromobenzyl aldehyde | PCN-201(Fe)-Cu | 82 |
| 9[b] | 4-Cyanobenzyl aldehyde | PCN-201(Fe)-Cu | 81 |
| 10[b] | 4-Nitrobenzyl aldehyde | PCN-201(Fe)-Cu | 68 |
| 11 | 2-Furyl aldehyde | PCN-201(Fe)-Cu | 93 |
| 12 | Thiophene 2-carboxaldehyde | PCN-201(Fe)-Cu | 82 |

[a] Reaction conditions: generally, aldehyde (1 mmol), aniline (1 mmol), TMSCN (1 mmol), and catalyst (0.1 mol% based on Cu or 0.05 mol% based on Fe) were placed in a 4 mL vial and stirred at room temperature for 10 min. Yields were determined by $^1$H-NMR analysis and calculated based on the ratios of product/(product + starting material)
[b] Reaction temperature: 50 °C

(Fe) has Lewis acidic Fe-porphyrin centers, while Cu-pyridyl moieties are absent. Neither the individual catalyst nor the combination of the two were found to display a catalytic activity anywhere close to that of PCN-201(Fe)-Cu, indicating the cooperative effect of two catalytically active sites (i.e., Cu-pyridyl and Fe-porphyrin sites). Indeed, previous research has shown that the formation of the α-aminonitrile requires the activation of both the imine group and TMSCN, each interact with one of the metal centers. PCN-201(Fe)-Cu represents an integration of high valence Fe$^{III}$ and low valence Cu$^I$ centers, which simultaneously activates the electrophiles and nucleophiles. It should be emphasized that the PCN-201(Fe)-Cu is not a simple mixture of the Cu and Fe MOF catalysts. The catalytic performance of a physical mixture of PCN-201(Ni)-Cu and PCN-224(Fe) is much lower than that of PCN-201(Fe)-Cu, which provides strong evidence to the bimetallic cooperative mechanism. For the physical mixture of PCN-201(Ni)-Cu and PCN-224 (Fe), reaction intermediates formed at Fe$^{III}$ center in PCN-224 (Fe) need to diffuse to the Cu$^I$ sites in PCN-201(Ni)-Cu to fulfill the reaction cycle, which slow down the reaction rate. The ordered framework structure of PCN-201(Fe)-Cu affords the periodic arrangement and prearranged proximity of the Fe$^{III}$ and Cu$^I$ centers within a cavity, resulting in a cooperative effect and improved catalytic activity.

Encouraged by these results, we explored the substrate scope of PCN-201(Fe)-Cu. As shown in Table 1, different aldehydes were converted to corresponding α-aminonitriles in good yields. A broad substrate scope of substrates bearing electron-rich and electron-deficient functional groups was tolerated. To show the heterogeneous nature of catalysis, hot filtration experiment was conducted by removing the MOF catalyst after 1 min and no further increase of yield was observed within 10 min. To evaluate the recyclability, PCN-201(Fe)-Cu catalyst was simply separated from the mixture at the end of the reaction by centrifugation, washed with acetone, and reused for the next round of reaction. The catalytic activity and crystallinity were well-maintained after three cycles (Supplementary Table 4 and Supplementary Figure 27). Based on the current work and findings in the literature, a reaction pathway for the Strecker reactions catalyzed by PCN-201(Fe)-Cu was proposed (Supplementary Figure 28)[48].

First, the aldehyde and amine readily undergo a condensation reaction to form the Schiff base, which could be further promoted by Lewis acidic Fe$^{III}$ species. The resulting Schiff base as electrophile was further activated by the Fe-porphyrin sites. Meanwhile, the Cu-pyridyl moiety activates the nucleophilic TMSCN by binding to the cyano group. Close proximity ensures the efficient contact of nucleophile and electrophile to form the product. Therefore, PCN-201(Fe)-Cu represents a rare example of cooperative bimetallic catalytic systems built in multi-component MOFs. The stepwise retrosynthesis of multi-component MOFs allowed the placement of different catalysts in a predetermined array within a solid state porous material, which enables the discovery of novel cooperative catalysts with an unprecedented degree of control.

In conclusion, we demonstrated a kinetic analysis method to guide the retrosynthesis of multi-component MOFs. Four complex MOFs with multiple metal–organic domains were constructed in a stepwise manner involving the sequential application of available modification reactions including linker installation and cluster incorporation under kinetic considerations. To demonstrate the utility of retrosynthesis, a corporative bimetallic catalyst was built in a MOF by the sequential incorporation of two collaborative metal sites. Although many challenges still remain in MOF retrosynthesis, especially in building a reaction library analogous to the large and diverse organic reaction library, the intrinsic designability of MOFs and the fast evolving MOF synthetic methodologies suggest that such a lofty goal is eventually achievable[52,53].

## Methods

**Synthesis of PCN-224**. PCN-224 was synthesized on the basis of previous reports with slight modifications[36]. ZrCl$_4$ (30 mg), H$_4$TCPP (10 mg), benzoic acid (600 mg), and DMF (3 mL) were charged in a Pyrex vial. The mixture was heated in a 120 °C oven for 24 h. After cooling to room temperature, cubic dark purple crystals of PCN-224 were harvested. The as-synthesized PCN-224 was washed thoroughly with DMF, water, and 1 M HCl in DMF to remove dangling ligands on Zr$_6$ clusters. Anal. calcd (%) for PCN-224: C, 44.04; H, 2.62; N, 4.28. Found: C, 50.34; H, 3.10; N, 3.64%.

**Synthesis of PCN-224(Ni)**. PCN-224(Ni) was synthesized by the metalation of PCN-224(no metal). The crystals of PCN-224 were incubated in the solution of Ni (NO$_3$)$_2$ at 100 °C for 24 h. After cooling to room temperature, cubic dark red

crystals of PCN-224(Ni) were collected and washed thoroughly with DMF (10 mg, yield: 70%). Alternatively, large single crystals of PCN-224(Ni) can be synthesized from Ni-TCPP. A DMF (3 mL) solution of $ZrCl_4$ (30 mg), Ni-TCPP (10 mg), and benzoic acid (600 mg) were charged in a Pyrex vial. The mixture was heated in a 120 °C oven for 24 h. After cooling to room temperature, cubic dark red crystals of PCN-224(Ni) were collected. As PCN-224(Ni) single crystals show strong X-ray diffraction, they were used for the structural characterization of the single crystal to single-crystal transformation process. Anal. calcd (%) for PCN-224(Ni): C, 42.22; H, 2.36; N, 4.10%. Found: C, 45.43; H, 3.13; N, 4.48%.

**Synthesis of PCN-224(Ni)-INA**. PCN-224(Ni) (10 mg), INA (30 mg), and DMF (4 mL) were charged in a Pyrex vial. The mixture was heated in an 80 °C oven for 24 h resulting in the coordination of INA to the $Zr_6$ cluster of PCN-224(Ni).

**Synthesis of PCN-201(Ni)-Cu**. After the synthesis of PCN-224(Ni)-INA, the supernatant was decanted, and CuI (30 mg) and MeCN (4 mL) were added. The mixture was heated in a 65 °C oven for 24 h to generate the crystals of PCN-201 (Ni)-Cu. The crystals of PCN-201(Ni)-Cu were collected by centrifugation and washed three times with aliquots of DMF. Anal. calcd (%) for PCN-201(Ni)-Cu: C, 39.75; H, 2.44; N, 5.15%. Found: C, 30.38; H, 3.11; N, 2.22%.

**Synthesis of PCN-201(Ni)-Ni**. After the synthesis of PCN-224(Ni)-INA, the supernatant was decanted and $NiCl_2$ (30 mg) and DMF (4 mL) were added. The mixture was heated in a 65 °C oven for 24 h to generate the crystals of PCN-201 (Ni)-Ni. The crystals of PCN-201(Ni)-Ni were collected by centrifugation and washed three times with aliquots of DMF. Anal. calcd (%) for PCN-201(Ni)-Ni: C, 42.10; H, 2.58; N, 5.45%. Found: C, 39.59; H, 3.58; N, 5.03%.

**Synthesis of PCN-224(Ni)-DCDPS**. PCN-224(Ni) (10 mg), DCDPS (30 mg), and DMF (4 mL) were charged in a Pyrex vial. The mixture was heated in an 80 °C oven for 24 h resulting in the coordination of DCDPS to the $Zr_6$ cluster of PCN-224(Ni).

**Synthesis of PCN-202(Ni)-Hf**. After the synthesis of PCN-224(Ni)-DCDPS, to the solution was added $HfCl_4$ (30 mg) and acetic acid (0.1 mL). The mixture was heated in an 80 °C oven for 24 h. The crystals of PCN-202(Ni)-Hf were collected by centrifugation and washed three times with aliquots of DMF. Anal. calcd (%) for PCN-202(Ni)-Hf: C, 44.00; H, 2.72; N, 2.28%. Found: C, 45.82; H, 3.45; N, 4.68%.

**Synthesis of PCN-202(Ni)-Zr**. After the synthesis of PCN-224(Ni)-DCDPS, to the solution was added $ZrCl_4$ (30 mg) and acetic acid (0.1 mL). The mixture was heated in an 80 °C oven for 24 h. The crystals of PCN-202(Ni)-Zr were collected by centrifugation and washed three times with fresh DMF. Anal. calcd (%) for PCN-202(Ni)-Zr: C, 45.61; H, 2.82; N, 2.36%. Found: C, 46.98; H, 3.50; N, 3.82%.

**Synthesis of PCN-201(Fe)-Cu**. The synthesis of PCN-201(Fe)-Cu is similar to that of PCN-201(Ni)-Cu except that $FeCl_2$ was used for the metallation of PCN-224. The black crystals of PCN-201(Fe)-Cu were collected by centrifugation and washed three times with aliquots of DMF. Anal. calcd (%) for PCN-201(Fe)-Cu: C, 39.15; H, 2.40; N, 5.07%. Found: C, 37.77; H, 3.48; N, 5.02%.

**Characterization**. Gas sorption measurements were conducted using a Micrometritics ASAP 2020 system. PXRD was carried out with a Bruker D8-Focus Bragg-Brentano X-ray powder diffractometer equipped with a Cu-sealed tube ($\lambda = 1.54178$ Å) at 40 kV and 40 mA. SC-XRD was measured on a Bruker Venture CMOS diffractometer equipped with a Cu-$K_\alpha$ sealed-tube X-ray source ($\lambda = 1.54184$ Å) or Mo-$K\alpha$ sealed-tube X-ray source ($\lambda = 0.71073$ Å). NMR data were collected on a Mercury 300 MHz spectrometer. UV−Vis absorption spectra were recorded on a Shimadzu UV-2450 spectrophotometer. ICP-MS data were collected with a Perkin Elmer NexION® 300D ICP-MS. TGA was conducted on a TGA-50 (SHIMADZU) thermogravimetric analyzer. Infrared (IR) measurements were performed on a SHIMADZU IR Affinity-1 spectrometer. Field-emission scanning electron microscopy images were collected on the FEI Quanta 600 field-emission scanning electron microscope (America) at 20 kV.

**Data availability**. The X-ray crystallographic data for structures reported in this article have been deposited at the Cambridge Crystallographic Data Centre (CCDC), under deposition number CCDC 1544108-1544113. These data can be obtained free of charge from The Cambridge Crystallographic Data Centre via www.ccdc.cam.ac.uk/data_request/cif. All relevant data supporting the findings of this study are available from the corresponding authors on request.

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

## Acknowledgements

The gas adsorption-desorption studies of this research were supported by the Center for Gas Separations Relevant to Clean Energy Technologies, an Energy Frontier Research Center funded by the US Department of Energy, Office of Science, Office of Basic Energy Sciences under award number DE-SC0001015. Structural analyses were supported as part of the Hydrogen and Fuel Cell Program under award number DE-EE0007049. This computational work was funded by the Robert A. Welch Foundation through a Welch Endowed Chair to HJZ (A-0030). Texas A&M Supercomputing Facility was acknowledged to provide computing resources. The Distinguished Scientist Fellowship Program (DSFP) at KSU is gratefully acknowledged for supporting this work. We acknowledge the financial support of US Department of Energy Office of Fossil Energy, National Energy Technology Laboratory (DE-FE0026472). S.Y. also acknowledges the Texas A&M Energy Institute Graduate Fellowship Funded by ConocoPhillips and Dow Chemical Graduate Fellowship. Professor Michael O'Keeffe (Arizona State University, USA) was acknowledged for his helpful comments on the structures.

## Author contributions

Original idea was conceived by H.-C.Z. and S.Y.; experiments and data analysis were performed by S.Y., J.-S.Q., J.L., L.F., Y.F., J.P., and L.Z.; structural characterization was performed by D.S.; molecular simulations were performed by L.H. and T.C.; manuscript was drafted by H.-C.Z, S.Y., J.-S.Q., C.L., and A.A. All authors have given approval to the manuscript.
