## [Peer Review File · Nature Communications]

Reviewers' comments:

Reviewer #1 (Remarks to the Author):

In this contribution, the authors introduced the concept "retrosynthesis" into MOF chemistry for the first time and synthesized a series of PCN-224-based multi-component MOF materials with isorecticular structures and outstanding water stability, while direct one-pot hydrothermal or solvothermal reactions often lead to undesired thermodynamically favored products. In addition, the PCN-201(Fe)-Cu exhibited cooperatively bimetallic catalysis activity towards the three-component Strecker reactions, and reached higher yields than a simple mixture of Cu- and Fe-containing materials. This work expanded the synthetic approaches for MOF chemistry, so I reckon it worth publishing on Nat. Commun. after addressing the following comments:

1. The concept "retrosynthesis" is not very clear to readers who are not familiar with total synthesis. There should be a brief description about "retrosynthesis" and its application in organic chemistry in the introduction part. It will also be helpful to provide a scheme of "retrosynthesis" containing one example of organic synthesis as well as this MOF study to show the similarities.
2. Line 203: Hf₆ clusters are significantly larger than Zr₆, but the synthesis of both PCN-202(Ni)-Hf and PCN-202(Ni)-Zr are successful. How can SBUs with a different size be incorporated via retrosynthesis? Authors need to give proper explanation.
3. Line 205-206: Authors claimed that OBC and CDC cannot be incorporated into PCN-202 due to the symmetry. To confirm this, a second example with C_{2v} symmetry other than DCDPS should be studied.
4. Line 297: The compositions were further confirmed by EDX, ICP-MS experiments. Besides composition of metals, elemental analysis is important to complete this study.
5. The authors should give a proper explanation why PCN-244(Fe)-Cu has a higher catalysis yield than a mixture of Cu- and Fe-containing MOFs.
6. In Table 1, the yields should have an error bar.
7. Three cycles are not enough to claim complete recyclability, so it is recommended to perform the reaction to more cycles.
8. In order to confirm the heterogeneity of the MOF catalyst, a hot filtration experiment as well as N₂ sorption of the MOF after catalysis should be given.
9. The crystallinity of PCN-201(Fe)-Cu decreases during catalysis. Does dissolution occur to some extent? ICP of the reaction supernatant should be provided.
10. The temperature of H₂ sorption is not given.

Reviewer #2 (Remarks to the Author):

This submission aims at reporting a new concept in the preparation of multi-component MOFs. The work describes post-synthetic consecutive modifications of a PCN204 sample resulting in the successful preparation of other MOFs containing up two additional ligands and an additional metal cluster, plus other metals forming complexes. Although establishing the concept of retrosynthesis in the preparation of MOFs will require many more examples, the present submission illustrates that, at least for the present case, the concept of starting with a MOF meeting three requirements of stability, large pore size and coordinatively unsaturated positions at the constitutional nodes can serve to prepare these multi-component MOFs with well established ordering. Publication in Nature Communications is recommended to trigger further development of the concept presented of retrosynthesis. The authors should provide a revised version addressing these, relatively minor, comments:

The catalytic part of this work needs to be completed and performed in a more competent way. First, time-conversion plots (rather than just the yield at 10 min) should be given. Probably the authors have to make the reaction slower, by decreasing the amount of catalyst. Important: the authors have to indicate if the NMR spectrum to quantify was done at 10 min or if the samples were left for some time before recording the NMR spectra used for quantification. This can mislead

the yields, since the reaction can progress further until the spectrum is recorded. How long does it take to record the NMR spectrum in comparison with the 10 min reaction time?

Second, the control using the mixture of two different powders should be allowed to go to completion. It is the initial reaction rates, rather than the conversion at 10 min, the figure of merit to be compared, but no rates are given in the manuscript.

Third, the authors have to do the leaching tests. On one hand the authors have to start the reaction under the typical reaction conditions and filtrate the catalyst when the conversion is between 20-40 %. Is any further progress of the reaction observed in the absence of solid? On the other hand, chemical analysis of the solution at final reaction times and the reused solid catalyst have to be performed to convince that, even though cyanide is a good ligand, no metal leaching takes place under the reaction conditions.

It is difficult in Figure 5 to see exactly the pore size of PCN201(Ni)-Cu. The exact value should be given. Is it the same for PCN(Fe)-Cu? Can the reaction take place inside the pores of modified multi-metal MOF?

The authors should indicate the conversion value at which the turnover frequency of 6000 h⁻¹ was measured.

When referring to the use of MOFs as heterogeneous catalysis (refs 12-15), besides references on gas adsorption and separation (ref. 13), on CO₂ capture (ref 15) or even a reference in specific use of homochiral MOFs (ref. 12), some general references on the use of MOFs as solid catalysts for liquid phase reactions such Chem. Rev. 2010, 110, 4606 and other general reviews should be given.

I think that the other parts of the manuscript are convincing, the preparation of the materials is sufficiently described and the modified MOFs convincingly characterized. Once the previous points on catalysis are solved, publication in Nat. Commun. should proceed.

Reviewer #3 (Remarks to the Author):

The manuscript by Zhou and co-workers describes the use of a strategy which they describe as 'retrosynthesis' to construct a multicomponent MOF. This work is sufficiently different in its approach to synthesizing multicomponent MOFs; recent examples include Li (J. Am. Chem. Soc. 139, 1778-1781 (2017) and Telfer (J. Am. Chem. Soc. 137, 3901-3909 (2015).). The current study advances the concept of using 'retro synthetic analysis' to design new MOF materials, however, the structural and chemical constraints associated with this approach limits its generality. In addition, very similar ideas have been posited by the same group (J. Am. Chem. Soc., 2016, 138, 8912) although they were not branded 'reterosynthetic'. Nevertheless, the paper is presented in a scholarly fashion and the materials are well characterised. In summary, the work clearly shows the potential for controlling MOF chemistry via multicomponent synthesis and advances the field enough to be considered for publication in Nature commun. Accordingly, I recommend publication in its current form.

Reviewer #4 (Remarks to the Author):

All the crystal structures currently contain significant issues, and cannot be considered publishable.

General comments:

The sentence "Based on statistics of systematic absence, the space group of Im-3m was the best choice with lowest CFOM factor, as it was well-transformed to get Fourier peaks by direct

methods." implies that the CFOM (which is determined in XPREP or APEX2) is dependent of the results of the structure solution. This is untrue, the CFOM value is determined independently of, and prior to, the structure solution.

The sentence "The contribution of this region to the total structure factor was calculated via a discrete Fourier transformation and subtracted in order to generate a new set of hkl reflections by means of the program SQUEEZE." is incorrect. This is how SQUEEZE functioned in the past, however now it works in a different way. SQUEEZE now produces an additional file (the .FAB file), which contains a fixed contribution to the electron density from the disordered solvent region. The structure is then refined by refining the combination of the structural model and this fixed contribution against the original data.

"During the refinement, we used SIMU and DFIX to make the Ueq and molecular geometry much more reasonable." - this does not include all the restraints used in the different structure refinements, and does not specify which restraints were used to restraint what values. A much more extensive discussion of any restraints used should be included. Please note that a discussion of any restraints used for a particular structure should also be included in the `_refine_special_details` section of each individual CIF.

For all structures any A- and B-level CIF check alerts should be explained (using the appropriate `_vrf's`) or rectified.

A number of the structures contain very short ($<1\text{\AA}$) Zr-Zr 'bonds'. The structures should be appropriately modelled using PART instructions to remove these 'bonds'. In some cases the connectivity lists also include Zr-C bonds to COO carbon atoms. These should also be removed, as they do not represent actual bonding interactions. Finally, the authors should consider whether the longer range Zr-Zr bonds are actually bonding interactions, and prune the connectivity list appropriately.

PCN201Cu

There is a number of missing data items in this CIF that should be present:

`_exptl_absorpt_correction_T_min` ?
`_exptl_absorpt_correction_T_max` ?
`_diffrn_source` ?
`_diffrn_measurement_method` ?
`_diffrn_detector_area_resol_mean` ?
`_computing_data_collection` ?
`_computing_cell_refinement` ?
`_computing_data_reduction` ?
`_computing_structure_solution` ?
`_computing_molecular_graphics` ?
`_computing_publication_material` ?
`_refine_special_details` ?
`_atom_sites_solution_primary` ?
`_atom_sites_solution_secondary` ?

This structure has 2 A- and 6 B-level CIF check alerts that should be explained or rectified. The authors should give consideration as to whether they should model additional disorder in this structure. A discussion of the disorder handling and any restraints used should be included in the `_refine_special_details` section of the CIF.

PCN-201Ni

There is a number of missing data items in this CIF that should be present:

_diffraction_source ?
_diffraction_measurement_method ?
_diffraction_detector_area_resol_mean ?
_computing_data_collection ?
_computing_cell_refinement ?
_computing_data_reduction ?
_computing_structure_solution ?
_computing_molecular_graphics ?
_computing_publication_material ?
_refine_special_details ?
_atom_sites_solution_primary ?
_atom_sites_solution_secondary ?

This structure has 3 A- and 8 B-level CIF check alerts that should be explained or rectified. The authors should give consideration as to whether they should model additional disorder in this structure. A discussion of the disorder handling and any restraints used should be included in the `_refine_special_details` section of the CIF. The authors are reminded that the ISOR restraint is only intended for use on isolated atoms, and is inappropriate for bonded atoms (DELU, RIGU and SIMU should be used for such atoms). This structure can be refined un-DAMPed if the EXTI instruction is removed - this value refines to 0 in any case.

PCN202Hf

There is a number of missing data items in this CIF that should be present:

_diffraction_source ?
_diffraction_measurement_method ?
_diffraction_detector_area_resol_mean ?
_computing_data_collection ?
_computing_cell_refinement ?
_computing_data_reduction ?
_computing_structure_solution ?
_computing_molecular_graphics ?
_computing_publication_material ?
_refine_special_details ?
_atom_sites_solution_primary ?
_atom_sites_solution_secondary ?

This structure contains a serious error. The largest Q-peak present in the difference map is a water molecule bridging between two Zr sites, which has not been modelled. Additionally, the geometry of one of the carboxylic groups is physically unrealistic - the C-C-O angle is 93 degrees, and the O-C-O angle is 173 degrees. A discussion of the disorder handling and any restraints used should be included in the `_refine_special_details` section of the CIF.

PCN202Zr

There is a number of missing data items in this CIF that should be present:

_diffraction_source ?

_diffraction_measurement_method ?
_diffraction_detector_area_resolution_mean ?
_computing_data_collection ?
_computing_cell_refinement ?
_computing_data_reduction ?
_computing_structure_solution ?
_computing_molecular_graphics ?
_computing_publication_material ?
_refine_special_details ?
_atom_sites_solution_primary ?
_atom_sites_solution_secondary ?

Disorder of the macrocycle system should be modelled. This structure has 6 A- and 7 B-level CIF check alerts that should be explained or rectified. A discussion of the disorder handling and any restraints used should be included in the `_refine_special_details` section of the CIF.

PCN224DCDPS

There is a number of missing data items in this CIF that should be present:

_diffraction_source ?
_diffraction_measurement_method ?
_diffraction_detector_area_resolution_mean ?
_computing_data_collection ?
_computing_cell_refinement ?
_computing_data_reduction ?
_computing_structure_solution ?
_computing_molecular_graphics ?
_computing_publication_material ?
_refine_special_details ?
_atom_sites_solution_primary ?
_atom_sites_solution_secondary ?

This structure has 5 A- and 11 B-level CIF check alerts that should be explained or rectified. The authors should give consideration as to whether they should model additional disorder in this structure. A discussion of the disorder handling and any restraints used should be included in the `_refine_special_details` section of the CIF.

PCN224INA

There is a number of missing data items in this CIF that should be present:

_diffraction_source ?
_diffraction_measurement_method ?
_diffraction_detector_area_resolution_mean ?
_computing_data_collection ?
_computing_cell_refinement ?
_computing_data_reduction ?
_computing_structure_solution ?
_computing_molecular_graphics ?
_computing_publication_material ?
_refine_special_details ?

_atom_sites_solution_primary ?
_atom_sites_solution_secondary ?

Disorder of the macrocycle system should be modelled. This structure has 5 A- and 8 B-level CIF check alerts that should be explained or rectified. A discussion of the disorder handling and any restraints used should be included in the `_refine_special_details` section of the CIF.

Response to Reviewers' comments:

Reviewer #1:

In this contribution, the authors introduced the concept "retrosynthesis" into MOF chemistry for the first time and synthesized a series of PCN-224-based multi-component MOF materials with isoreticular structures and outstanding water stability, while direct one-pot hydrothermal or solvothermal reactions often lead to undesired thermodynamically favored products. In addition, the PCN-201(Fe)-Cu exhibited cooperatively bimetallic catalysis activity towards the three-component Strecker reactions, and reached higher yields than a simple mixture of Cu- and Fe-containing materials. This work expanded the synthetic approaches for MOF chemistry, so I reckon it worth publishing on *Nat. Commun.* after addressing the following comments:

Response: We appreciate your supportive comments.

1. The concept "retrosynthesis" is not very clear to readers who are not familiar with total synthesis. There should be a brief description about "retrosynthesis" and its application in organic chemistry in the introduction part. It will also be helpful to provide a scheme of "retrosynthesis" containing one example of organic synthesis as well as this MOF study to show the similarities.

Response: Thank you for the suggestion. We have explained the "retrosynthesis" and its application before the discussion of "MOF retrosynthesis". The following discussions have been added in the Introduction section of revised manuscript. Figure 1 has also been added as a schematic representation to show the similarity between "retrosynthesis" of organic molecules and MOFs.

"Retrosynthetic analysis is a concept to approach the synthesis of complex organic molecules by transforming a target molecule into simpler precursors and sequentially applying a set of known chemical reactions to assemble them together."

2. Line 203: Hf₆ clusters are significantly larger than Zr₆, but the synthesis of both PCN-202(Ni)-Hf and PCN-202(Ni)-Zr are successful. How can SBUs with a different size be incorporated via retrosynthesis? Authors need to give proper explanation.

Response: The atomic radius of Zr (1.55 Å) and Hf (1.55 Å) are very close because of the lanthanide contraction. Therefore, the Hf₆ clusters and Zr₆ clusters are expected to have similar size, so that they can form isostructural PCN-202(Ni)-Hf and PCN-202(Ni)-Zr.

3. Line 205-206: Authors claimed that OBC and CDC cannot be incorporated into PCN-202 due to the symmetry. To confirm this, a second example with C_{2v} symmetry other than DCDPS should be studied.

Response: Thank you for the comments. The symmetry of the linker is not the criteria for their successful incorporation. The position and direction of the carboxylates determines whether the linker is suitable for the PCN-202 or not. Although CDC has the same symmetry as DCDPS, the dihedral angle between the carboxylate group and the equatorial plane (0° for CDC and 90° for DCDPS) is not suitable for the PCN-202 structure. In addition, the size of the linker need to fit in the cavity of PCN-202. Take these requirements into consideration, DCDPS is the only suitable ligand we can find. The following discussions have been modified in the revised manuscript.

“Two other linkers, OBC and CDC, with C_2 and C_{2v} symmetry respectively, have also been attempted to construct PCN-202 structures (OBC = 4,4'-oxybisbenzoate, CDC = 3,6-carbazoledicarboxylate). Although they have approximately the same size with DCDPS, the dihedral angles between the carboxylate group and the equatorial plane ($\sim 40^\circ$ for OBC and 0° for CDC) are not suitable for the PCN-202 structure. Therefore, they did not give rise to PCN-202 under identical synthetic condition.”

4. Line 297: The compositions were further confirmed by EDX, ICP-MS experiments. Besides composition of metals, elemental analysis is important to complete this study.

Response: Element analyses were conducted. The elemental analysis results are comparable to the calculations based on their formulae. These data have been added and highlighted in the revised manuscript.

5. The authors should give a proper explanation why PCN-244(Fe)-Cu has a higher catalysis yield than a mixture of Cu- and Fe-containing MOFs.

Response: For the physical mixture of PCN-201(Ni)-Cu and PCN-224(Fe), reaction intermediates formed at Fe^{III} center in PCN-224(Fe) need to diffuse to the Cu^I sites in PCN-201(Ni)-Cu to fulfill the reaction cycle, which slow down the reaction rate. The ordered framework structure of PCN-201(Fe)-Cu affords the periodic arrangement and prearranged proximity of the Fe^{III} and Cu^I centers within a cavity, resulting in a cooperative effect and improved catalytic activity. These discussions have been added in the revised manuscript.

6. In Table 1, the yields should have an error bar.

Response: We agree that error bar is needed to show the accuracy and reproducibility of the data. This is especially important when we compare the performance of different catalysts. The yield catalyzed by different catalysts were repeated three times and the standard error has been added in Supplementary Table 4.

7. Three cycles are not enough to claim complete recyclability, so it is recommended to perform the reaction to more cycles.

Response: To show the recyclability, we performed the recycle experiment for 5 cycles with a maintained yield of 90 %. The slightly reduced yield is likely to be attributed to the loss of MOF particles during the centrifugation.

8. In order to confirm the heterogeneity of the MOF catalyst, a hot filtration experiment as well as N_2 sorption of the MOF after catalysis should be given.

Response: Hot filtration experiment was conducted by removing the MOF catalyst after 1 min when the yield of aminonitriles reached 35 %. No further increase of yield was observed within 10 min. 1H NMR analysis indicated that the reaction stopped at the imine intermediate (yield of 47 %). This result suggested that MOF as a catalyst was required to further convert imine intermediate into the final product. For comparison, the yield of aminonitriles reached 99 % within 10 min at the existence of MOF catalyst. These results corroborated the true heterogeneous nature of catalysis.

The N₂ sorption isotherms of PCN-201(Fe)-Cu after the catalytic reaction were further tested to show the maintained porosity (Supplementary Figure 27).

9. The crystallinity of PCN-201(Fe)-Cu decreases during catalysis. Does dissolution occur to some extent? ICP of the reaction supernatant should be provided.

Response: The slight peak broaden of PXRD is attributed to the reduced particle size by stirring. In addition, different amounts of samples were used for PXRD measurements which explains the different diffraction intensities. No Fe, or Zr species were observed in the reaction supernatant by ICP measurements suggesting that the MOF was intact during catalysis. A small amount of Cu (~0.1 μmol, corresponding to ~10 % of total Cu in catalyst) was detected in the reaction supernatant which is caused by the removal of uncoordinated Cu species in the pore. ¹H NMR analysis eliminate the existence of INA or TCPP in the reaction supernatant, further substantiate the stability of MOF during catalysis.

10. The temperature of H₂ sorption is not given.

Response: Thank you for pointing it out. The H₂ sorption measurements were conducted at 77 K. This information has been added in Supplementary Figures 21 and 22.

Reviewer #2:

This submission aims at reporting a new concept in the preparation of multi-component MOFs. The work describes post-synthetic consecutive modifications of a PCN204 sample resulting in the successful preparation of other MOFs containing up to two additional ligands and an additional metal cluster, plus other metals forming complexes. Although establishing the concept of retrosynthesis in the preparation of MOFs will require many more examples, the present submission illustrates that, at least for the present case, the concept of starting with a MOF meeting three requirements of stability, large pore size and coordinatively unsaturated positions at the constitutional nodes can serve to prepare these multi-component MOFs with well-established ordering. Publication in Nature Communications is recommended to trigger further development of the concept presented of retrosynthesis. The authors should provide a revised version addressing these, relatively minor, comments:

Response: We appreciate your supportive comments.

The catalytic part of this work needs to be completed and performed in a more competent way. First, time-conversion plots (rather than just the yield at 10 min) should be given. Probably the authors have to make the reaction slower, by decreasing the amount of catalyst.

Response: We attempted to study the reaction time-course but to no avail, because of the fast reaction rate. MOF catalyst was removed after 1, 5, and 10 min respectively and the yield analyzed immediately by ¹H NMR. The yields of aminonitriles were calculated to be 35 % (1 min), 91 % (5 min) and 99 % (10 min). Considering the fast reaction rate and the time needed to separate the catalyst, the yield after 1 min and 5 min can be overestimated. But it can be concluded that the reaction is finished before 10 min. It is difficult to control the reaction rate by reducing the amount of catalyst since only 0.1 mol% (~1 mg) MOF catalyst was used for each trial. Further decreasing the amount of catalyst will inevitably cause the inaccuracy in weight measurement.

Important: the authors have to indicate if the NMR spectrum to quantify was done at 10 min or if the samples were left for some time before recording the NMR spectra used for quantification. This can mislead the yields, since the reaction can progress further until the spectrum is recorded. How long does it take to record the NMR spectrum in comparison with the 10 min reaction time?

Response: The MOF catalyst was immediately separated from the reaction system after 10 min. About 100 μL of reaction supernatant was sampled, dissolved in 1 mL of dimethyl sulfoxide- d_6 and then measured by ^1H NMR. It took around 20 min before the spectrum was recorded. However, hot filtration experiments after 1 min indicated that the yield of aminonitrile (35 %) didn't increase after the removal of MOF catalyst. ^1H NMR analysis indicated that the reaction stopped at the imine intermediate without the MOF. MOF as a catalyst is required to further convert imine intermediate into the aminonitrile product. Therefore, the result of ^1H NMR can accurately represent the yield within 10 min reaction time.

Second, the control using the mixture of two different powders should be allowed to go to completion. It is the initial reaction rates, rather than the conversion at 10 min, the figure of merit to be compared, but no rates are given in the manuscript.

Response: The reaction time was extended to 1 h using the mixture of PCN-201(Ni)-Cu and PCN-224(Fe) to achieve a 99 % yield. We agree that the initial reaction rates are figure of merit to evaluate the catalytic activity. However, as aforementioned, the reaction kinetics of this system are difficult to measure and investigate because of the fast reaction rate. This problem has been encountered in other MOF based catalytic system as well, where yields at a certain time was used to estimate the reaction rate (*J. Am. Chem. Soc.* **2015**, *137*, 6132; *J. Am. Chem. Soc.* **2017**, *139*, 12382). Therefore, we use the yield at 10 min to compare the reaction rate by different catalysts.

Third, the authors have to do the leaching tests. On one hand the authors have to start the reaction under the typical reaction conditions and filtrate the catalyst when the conversion is between 20-40 %. Is any further progress of the reaction observed in the absence of solid? On the other hand, chemical analysis of the solution at final reaction times and the reused solid catalyst have to be performed to convince that, even though cyanide is a good ligand, no metal leaching takes place under the reaction conditions.

Response: Hot filtration experiments were conducted by removing the MOF catalyst after 1 min when the yield of aminonitriles reached 35 %. No further increase of yield was observed within 10 min. For comparison, the yield of aminonitriles reached 99 % within 10 min at the existence of MOF catalyst. These results corroborated the true heterogeneous nature of catalysis. No Fe, or Zr species were observed in the reaction supernatant by ICP measurements suggesting that the MOF is intact during catalysis. A small amount of Cu ($\sim 0.1 \mu\text{mol}$, corresponding to $\sim 10\%$ of total Cu in catalyst) was observed which is caused by the removal of uncoordinated Cu species in the cavity. ^1H NMR analysis eliminate the existence of INA or TCPP in the reaction supernatant, further substantiate the stability of MOF.

It is difficult in Figure 5 to see exactly the pore size of PCN-201(Ni)-Cu. The exact value should be given. Is it the same for PCN-201(Fe)-Cu? Can the reaction take place inside the pores of modified multi-metal MOF?

Response: Changing the metal center of the porphyrin linker is not expected to affect the pore size of the MOF (*J. Am. Chem. Soc.*, **2013**, *135*, 17105; *Angew. Chem. Int. Ed.*, **2012**, *51*, 10307). Although the single crystal structure of PCN-201(Fe)-Cu was not obtained, the pore size of PCN-201(Fe)-Cu is estimated to be 16 Å based on the crystal structure and pore size distribution of PCN-201(Ni)-Cu. The size of the cavity is large enough for the diffusion of the substrate (diameter < 6 Å) and the product (diameter < 11 Å).

The authors should indicate the conversion value at which the turnover frequency of 6000 h⁻¹ was measured.

Response: The turnover frequency of 6000 h⁻¹ were calculated based on the 99 % yield of 1 mmol product within 10 min using 0.1 mol% catalyst. And the relevant information has been added in the revised manuscript.

When referring to the use of MOFs as heterogeneous catalysis (refs 12-15), besides references on gas adsorption and separation (ref. 13), on CO₂ capture (ref 15) or even a reference in specific use of homochiral MOFs (ref. 12), some general references on the use of MOFs as solid catalysts for liquid phase reactions such Chem. Rev. 2010, 110, 4606 and other general reviews should be given.

Response: Thank you. The suggested references have been added in the revised manuscript.

I think that the other parts of the manuscript are convincing, the preparation of the materials is sufficiently described and the modified MOFs convincingly characterized. Once the previous points on catalysis are solved, publication in *Nat. Commun.* should proceed.

Reviewer #3:

The manuscript by Zhou and co-workers describes the use of a strategy which they describe as 'retrosynthesis' to construct a multicomponent MOF. This work is sufficiently different in its approach to synthesizing multicomponent MOFs; recent examples include Li (*J. Am. Chem. Soc.* 139, 1778-1781 (2017) and Telfer (*J. Am. Chem. Soc.* 137, 3901-3909 (2015)). The current study advances the concept of using 'retro synthetic analysis' to design new MOF materials, however, the structural and chemical constraints associated with this approach limits its generality. In addition, very similar ideas have been posited by the same group (*J. Am. Chem. Soc.*, 2016, 138, 8912) although they were not branded "reterosynthetic". Nevertheless, the paper is presented in a scholarly fashion and the materials are well characterised. In summary, the work clearly shows the potential for controlling MOF chemistry via multicomponent synthesis and advances the field enough to be considered for publication in Nature commun. Accordingly, I recommend publication in its current form.

Response: Thank you very much for your supportive comments.

Reviewer #4:

The sentence "Based on statistics of systematic absence, the space group of Im-3m was the best choice with lowest CFOM factor, as it was well-transformed to get Fourier peaks by direct methods." implies that the CFOM (which is determined in XPREP or APEX2) is dependant of the results of the structure solution. This is untrue, the CFOM value is determined independantly of, and prior to, the structure solution.

Response: We appreciate the reviewer's comments. This description is misleading so we deleted it. This section was also revised correspondingly.

The sentence "The contribution of this region to the total structure factor was calculated via a discrete Fourier transformation and subtracted in order to generate a new set of hkl reflections by means of the program SQUEEZE." is incorrect. This is how SQUEEZE functioned in the past, however now it works in a different way. SQUEEZE now produces an additional file (the .FAB file), which contains a fixed contribution to the electron density from the disordered solvent region. The structure is then refined by refining the combination of the structural model and this fixed contribution against the original data.

Response: Thanks for your constructive suggestions. You are right, based on ref. *Acta Cryst. C* 71, 3–8, (2015), the new treatment for disordered solvents using SQUEEZE will need to combine the .FAB file into final .CIF file. We changed the description about this to 'Platon SQUEEZE³ was used and .fab file was created containing partial structure factors representing the SQUEEZE region. The appropriate partial structure factors were used for input to SHELXL with the ABIN instruction. The ABIN instruction reads h, k, l, A and B from the file name fab, where A and B are the real and imaginary components of a partial structure factor⁴' We also used the generated .FAB to get new .CIF files for all structures, which now have been re-deposited into CCDC database.

"During the refinement, we used SIMU and DFIX to make the Ueq and molecular geometry much more reasonable." - this does not include all the restraints used in the different structure refinements, and does not specify which restraints were used to restraint what values. A much more extensive discussion of any restraints used should be included. Please note that a discussion of any restraints used for a particular structure should also be included in the `_refine_special_details` section of each individual CIF. For all structures any A- and B-level CIF check alerts should be explained (using the appropriate `_vrf`'s) or rectified.

Response: Thanks for your constructive suggestions. We added the refinement details for each structure in cif as well as SI materials as follows:

'For PCN201Cu, due to the weak diffractions at high Bragg angle, several atoms show unusual isotropic thermal parameters, thus several restraints were applied to ensure a reasonable refinement.

In details, SIMU restraints were also used for organic ligands and partial metal center with large thermal motion (O2 C9 C10 C12 N2 C11 C1 C2 O1 C4 C3 C6 C5 C7 C8 N1 Ni1). FLAT was used to ensure the planarity of some part of ligand. DFIX was used to fix the C9-C10 to 1.55 %A, C10-C11 and C11-C12 to 1.35 %A.

For PCN201Ni, due to the weak diffractions at high Bragg angle, several atoms show unusual isotropic thermal parameters, thus several restraints were applied to ensure a reasonable refinement.

In details, SIMU restraints were also used for organic ligands with large thermal motion (O3 O4 O1 O2 C9 C10 C11 N2 C12 O1 C1 C2 C3 C4 C5 C6 C8 C7 N1). FLAT was used to ensure the planarity of some part of ligand (O2 C9 C10 C11 N2 C12). DFIX was used to fix the C12-C11, C0-C11 and N2-C12 to 1.35 %A, N2-C11 and C10-C12 to 2.35 %A, C9-C10 to 1.55 %A.

For PCN202Hf, due to the weak diffractions at high Bragg angle, several atoms show unusual isotropic thermal parameters, thus several restraints were applied to ensure a reasonable refinement. In details, SIMU restraints were also used for organic ligands with large thermal motion (O1 C1 C3 C2 C4 C6 C7 C8 C5 N4 O3 C9 C10 C11 C12 C13 S4 C14 C15 C16 C17 C18 O5). FLAT was used to ensure the planarity of some part of ligand (C10 C11 C12 C13 C9 and C13 C12_1 C11_1 C10 C11 C12 C9 S4, EQIV 1 +X,+Z,+Y). DFIX was used to fix the C1-O1, O5-C18, O3-C9 to 1.25 Å; C14-C15, C15-C16, C16-C17, C11-C12, C12-C13, C10-C11 to 1.35 Å; C15-C17, C14-C16, C11-C13, C10-C12 to 2.35 Å; C17-C18 and C9-C10 to 1.55 Å; C14-S4, C13-S4 to 1.85 Å; C10-O3 to 2.5 Å.

For PCN202Zr, due to the weak diffractions at high Bragg angle and not good crystal quality, several atoms show unusual isotropic thermal parameters, thus several restraints were applied to ensure a reasonable refinement. In details, SIMU restraints were also used for organic ligands with large thermal motion (S1 O1 O4 C9 > C18). FLAT was used to ensure the planarity of some part of ligand (S1 C9 C10 C11 C12 C13). DFIX was used to fix the C14-C15, C13-C12, C11-C12, C10-C11 and C9-C10 to 1.35 Å, C13-C11 and C10-C12 to 2.35 Å, S1-C12 to 2.70 Å.

For PCN224DCDPS, due to the weak diffractions at high Bragg angle and not good crystal quality, several atoms show unusual isotropic thermal parameters, thus several restraints were applied to ensure a reasonable refinement. In details, SIMU restraints were also used for organic ligands with large thermal motion (O4 S1 C10 C11 C12 C13 O5 C9). FLAT was used to ensure the planarity of some part of ligand (C10 C11 C12 C13 C9). DFIX was used to fix the C13-C12, C11-C12, C10-C11 to 1.35 Å, C9-C10 to 1.55 Å, C13-C11 and C10-C12 to 2.35 Å, C13-S1 to 1.9 Å, S1-O5 to 1.45 Å; O5-C13 to 2.6 Å.

For PCN224INA, due to the weak diffractions at high Bragg angle, several atoms show unusual isotropic thermal parameters, thus several restraints were applied to ensure a reasonable refinement. In details, SIMU restraints were also used for organic ligands with large thermal motion (N2 C12 C11 C10 C9 O2). FLAT was used to ensure the planarity of some part of ligand (C10 N2 C12 C11). DFIX was used to fix the C12-C11, C10-C11, N2-C12 to 1.35 Å, N2-C11 and C10-C12 to 2.35 Å.'

A number of the structures contain very short (<1Å) Zr-Zr 'bonds'. The structures should be appropriately modelled using PART instructions to remove these 'bonds'. In some cases the connectivity lists also include Zr-C bonds to COO carbon atoms. These should also be removed, as they do not represent actual bonding interactions. Finally, the authors should consider whether the longer range Zr-Zr bonds are actually bonding interactions, and prune the connectivity list appropriately.

Response: Thanks for your constructive suggestions. We used 'part' instructions to treat disordered Zr atom and the short Zr-Zr 'bonds' have been eliminated. We also checked other kinds of 'bond' in the cif such as Zr-C and Zr-Zr and deleted these unreal bonding between them from cif file.

PCN201Cu: There is a number of missing data items in this CIF that should be present. This structure has 2 A- and 6 B-level CIF check alerts that should be explained or rectified. The authors should give consideration as to whether they should model additional disorder in this structure. A

discussion of the disorder handling and any restraints used should be included in the `_refine_special_details` section of the CIF.

Response: Thanks for your suggestions. We have added all above data loop in cif. A and B alters were treated by re-refinement or response using Validation Reply Form at the end of cif.

PCN-201Ni: There is a number of missing data items in this CIF that should be present. This structure has 3 A- and 8 B-level CIF check alerts that should be explained or rectified. The authors should give consideration as to whether they should model additional disorder in this structure. A discussion of the disorder handling and any restraints used should be included in the `_refine_special_details` section of the CIF. The authors are reminded that the ISOR restraint is only intended for use on isolated atoms, and is inappropriate for bonded atoms (DELU, RIGU and SIMU should be used for such atoms). This structure can be refined un-DAMPed if the EXTI instruction is removed - this value refines to 0 in any case.

Response: Thanks for your suggestions. We have added all above data loop in cif. We removed the EXTI instruction in ins file and damp instruction was also deleted to make the refinement to be convergent. The ISOR was also removed in ins file. In all, both A and B alters were treated by re-refinement or answered using Validation Reply Form at the end of cif. We added the refinement details for each structure in cif as well as SI materials.

PCN202Hf: There is a number of missing data items in this CIF that should be present. This structure contains a serious error. The largest Q-peak present in the difference map is a water molecule bridging between two Zr sites, which has not been modelled. Additionally, the geometry of one of the carboxylic groups is physically unrealistic - the C-C-O angle is 93 degrees, and the O-C-O angle is 173 degrees. A discussion of the disorder handling and any restraints used should be included in the `_refine_special_details` section of the CIF.

Response: Thanks for your suggestions. We have added all above data loop in cif. We assigned Q1 to one water molecule O4 and re-refined structure. The R_1 was also reduced obviously. Using the DFIX constraints, the O-C-O angle become more reasonable as 132°, 125° and 127°, respectively. In all, both A and B alters were treated by re-refinement or answered using Validation Reply Form at the end of cif. We added the refinement details for this structure in cif as well as SI materials.

PCN202Zr: There is a number of missing data items in this CIF that should be present. Disorder of the macrocyle system should be modelled. This structure has 6 A- and 7 B-level CIF check alerts that should be explained or rectified. A discussion of the disorder handling and any restraints used should be included in the `_refine_special_details` section of the CIF.

Response: Thanks for your suggestions. We have added all above data loop in cif. We have tried our best to refine the macrocyle system in a disorder fashion, but failed finally, which may be caused by the weak diffractions at higher Bragg angles. All alerts were treated by re-refinement or answered using Validation Reply Form at the end of cif. We added the refinement details for this structure in cif as well as SI materials.

PCN224DCDPS: There is a number of missing data items in this CIF that should be present. This structure has 5 A- and 11 B-level CIF check alerts that should be explained or rectified. The authors should give consideration as to whether they should model additional disorder in this structure. A

discussion of the disorder handling and any restraints used should be included in the `_refine_special_details` section of the CIF.

Response: Thanks for your constructive suggestions. We have added all above data loop in cif. We have tried our best to refine the disordered part in this structure, but failed finally, which may be caused by the weak diffractions at higher Bragg angles and the highly dispersed electron clouds. All alerts were treated by re-refinement or answered using Validation Reply Form at the end of cif. We added the refinement details for this structure in cif as well as SI materials.

PCN224INA: There is a number of missing data items in this CIF that should be present. Disorder of the macrocycle system should be modelled. This structure has 5 A- and 8 B-level CIF check alerts that should be explained or rectified. A discussion of the disorder handling and any restraints used should be included in the `_refine_special_details` section of the CIF.

Response: Thanks for your constructive suggestions. We have added all above data loop in cif. We have tried our best to refine the macrocycle system in a disorder fashion, but failed finally, which may be caused by the weak diffractions at higher Bragg angles. All alerts were treated by re-refinement or answered using Validation Reply Form at the end of cif. We added the refinement details for this structure in cif as well as SI materials.

REVIEWERS' COMMENTS:

Reviewer #1 (Remarks to the Author):

The authors have addressed my comments and made the needed changes correctly. I am pleased to recommend the acceptance of the paper in this version now.

Reviewer #2 (Remarks to the Author):

The authors have taken most of the points of my report in a constructive way and have performed most of the required controls. I still think that it is always possible to slow down a catalytic reaction and that comparison of the temporal profiles is necessary when studying kinetics. If the authors cannot diminish the amount of catalyst, they could have add ten times more substrates and reagents, etc. On the other hand, 10 % Cu leaching is not negligible and it indicates the lack of catalytic stability at long term. I would have liked to see additional Cu analysis in other reuses. In any case, in general terms I am satisfied and publication in Nature Communications is recommended.

Reviewer #5 (Remarks to the Author):

The crystallography of MOF's such as the one presented here is difficult at best. Reviewers #4 had a number of comments which the authors have address in full. I believe the crystallography here has now be done as well as possible base on the data.

Response to Reviewers' comments:

Reviewer #1:

The authors have addressed my comments and made the needed changes correctly. I am pleased to recommend the acceptance of the paper in this version now.

Response: Thank you for your supportive comments.

Reviewer #2:

The authors have taken most of the points of my report in a constructive way and have performed most of the required controls. I still think that it is always possible to slow down a catalytic reaction and that comparison of the temporal profiles is necessary when studying kinetics. If the authors cannot diminish the amount of catalyst, they could have add ten times more substrates and reagents, etc. On the other hand, 10 % Cu leaching is not negligible and it indicates the lack of catalytic stability at long term. I would have liked to see additional Cu analysis in other reuses. In any case, I general terms I am satisfied and publication in Nature Communications is recommended.

Response: Thank you very much for the suggestions. We agree that the kinetic studies are important. We attempted to study the reaction time-course by removing the MOF catalyst after 1, 5, and 10 min respectively. The yields of aminonitriles were calculated to be 35 % (1 min), 91 % (5 min) and 99 % (10 min). More detailed kinetic studies were unsuccessful because of the fast reaction rate. We do observe 10 % Cu leaching during cycling experiment. Recycle experiment for 5 cycles shows a yield of 90 % which demonstrates the reusability of the catalyst.

Reviewer #5:

The crystallography of MOF's such as the one presented here is difficult at best. Reviewers #4 had a number of comments which the authors have address in full. I believe the crystallography here has now be done as well as possible base on the data.

Response: We appreciate your supportive comments.